# The Phase Diagram of the API Benzocaine and Its Highly Persistent, Metastable Crystalline Polymorphs

**DOI:** 10.3390/pharmaceutics15051549

**Published:** 2023-05-20

**Authors:** Ivo B. Rietveld, Hiroshi Akiba, Osamu Yamamuro, Maria Barrio, René Céolin, Josep-Lluís Tamarit

**Affiliations:** 1Institute for Solid State Physics, University of Tokyo, 5-1-5 Kashiwanoha, Kashiwa 217-8581, Chiba, Japan; 2Université Rouen Normandie, SMS, UR 3233, F-76000 Rouen, France; 3Faculté de Pharmacie, Université Paris Cité, F-75006 Paris, France; 4Group de Caracterizació de Materials, Departament de Fisica, EEBE, Universitat Politècnica de Catalunya, Eduard Maristany, 10-14, 08019 Barcelona, Catalonia, Spainjosep.lluis.tamarit@upc.edu (J.-L.T.); 5Barcelona Research Center in Multiscale Science and Engineering, Universitat Politècnica de Catalunya, Eduard Maristany, 10-14, 08019 Barcelona, Catalonia, Spain

**Keywords:** active pharmaceutical ingredient, phase behaviour, pressure–temperature phase diagram, thermodynamics, crystal structure, thermal expansion, adiabatic calorimetry

## Abstract

The availability of sufficient amounts of form I of benzocaine has led to the investigation of its phase relationships with the other two existing forms, II and III, using adiabatic calorimetry, powder X-ray diffraction, and high-pressure differential thermal analysis. The latter two forms were known to have an enantiotropic phase relationship in which form III is stable at low-temperatures and high-pressures, while form II is stable at room temperature with respect to form III. Using adiabatic calorimetry data, it can be concluded, that form I is the stable low-temperature, high-pressure form, which also happens to be the most stable form at room temperature; however, due to its persistence at room temperature, form II is still the most convenient polymorph to use in formulations. Form III presents a case of overall monotropy and does not possess any stability domain in the pressure–temperature phase diagram. Heat capacity data for benzocaine have been obtained by adiabatic calorimetry from 11 K to 369 K above its melting point, which can be used to compare to results from in silico crystal structure prediction.

## 1. Introduction

A pressure–temperature phase diagram involving the solid phases II and III of benzocaine has been reported previously [1]. The existence of a third polymorph, form I, was known at the time, but it had not been obtained during the recrystallisation experiments, and thus, the phase diagram containing exclusively the forms II and III had been constructed [1]. The II–III dimorphism represents a case of enantiotropy turning into monotropy at higher pressures, where form III is the more stable form. Enantiotropy implies that a stable equilibrium exists between two crystalline polymorphs and that by passing the equilibrium temperature at a given pressure, one polymorph will reversibly convert into the other. Monotropy indicates that one of the polymorphs is fully metastable, and thus only the transition from the metastable into the stable polymorph can be observed, whereas the reverse will not occur by a simple change in temperature. The terminology has been introduced by Lehmann based on microscopy studies with a heating stage [2]. Subsequently, Nagasako extended these notions by taking pressure into account, allowing for polymorphic systems to change their enantiotropic/monotropic behaviour as a function of pressure [3]. One of the earliest-established and better-known systems studied under pressure and temperature is that of sulphur, mentioned by Bakhuis Roozeboom [4]. It represents enantiotropy turning into monotropy with increasing pressure, which is therefore called sulphur-type phase behaviour (Figure 1). The three other possible configurations for the phase behaviour of two polymorphs are monotropy turning enantiotropic upon increasing the pressure (the inverse of the sulphur case), overall enantiotropy (both forms maintain a domain irrespective of the pressure) and overall monotropy (one form does not possess a stable domain as a function of pressure and temperature) [4,5].

As previously published [1], benzocaine forms II and III possess a similar phase relationship as sulphur, in which rhombic is replaced by form III and monoclinic by form II (Figure 1). The availability of benzocaine form I makes it possible to demonstrate how its phase relationships can be incorporated in the existing pressure–temperature phase diagram and that its incorporation does not affect the phase relationships between phases II and III.

To verify and improve the accuracy of in silico crystal structure prediction, heat capacities of different polymorphs from 0 K to room temperature are of interest, so that their calculated stability hierarchies and energy contents can be compared with experimental values [6,7,8,9]. Therefore, in this paper, heat capacity data are reported starting at 10 K (with extrapolation to 0 K) for forms I and III, which both convert into form II at higher temperatures.

The absence of form I in the formerly reported phase diagram [1] highlights the fact that polymorph II, once formed, does not convert back into form I and neither, it seems, does form III; however, as the previous paper on benzocaine demonstrates [1], forms II and III readily interconvert. This makes benzocaine a system of interest to study the persistence of metastable forms to help increase the solubility of a drug by using metastable forms [10,11,12,13].

Four different crystal structures of benzocaine have been solved and can be found in the Cambridge Structural Database, although the fourth polymorph, form IV, is only stable under pressure at room temperature [14]. In Table 1, the crystal structures obtained under ordinary pressure (i.e., the vapour pressure of benzocaine as a function of the temperature and no applied hydrostatic pressure) have been listed together with a single example of form IV at 0.55 GPa. Form I crystallises in the space group *P*2_1_/c with Z = 4, form II in *P*2_1_2_1_2_1_ with Z = 4, and form III in space group *P*2_1_ with Z = 8. Form IV (*P*2_1_/c, Z = 4) appears if form I at room temperature is subjected to a pressure of about 0.50 GPa, while form I reappears at 0.41 GPa with decreasing the pressure.

In a previous paper on the benzocaine phase diagram involving forms II and III [1], the following calorimetric data have been reported: the temperature of fusion (L is liquid) of form II, *T*_II→L_ = 362.4(5) K, and the enthalpy of fusion of form II, ∆_II→L_*H* = 141(3) J g^−1^, the equilibrium temperature between forms III and II, *T*_III→II_ = 265.3(5) K, and the enthalpy difference between forms III and II, ∆_III→II_*H* = 3.0(1.0) J g^−1^, for the transition from form III to form II upon heating, while the reverting transition on cooling from form II to form III occurs about 5 degrees lower, at 260 K [1]. Moreover, the pressure–temperature data that have been obtained by high-pressure thermal analysis are those for the III–II transition and the melting transition, which pass through the transition temperatures (*P* ≈ 0 MPa) mentioned just above [1]:*P*_III-II_(*T*)/MPa = −722(18) + 2.70(6) *T*/K (1)
*P*_II-L_(*T*)/MPa = −2166(50) + 5.99(13) *T*/K (2)

In general, commercial batches contain form II; however, form I can be obtained by recrystallisation from ethanol (Figure 2). The stability relationships between form I and the other two forms II and III have been investigated by adiabatic calorimetry, high-pressure differential thermal analysis and powder X-ray diffraction. With the help of the topological method [1,21,22,23], a full pressure–temperature phase diagram of the trimorphism of benzocaine is presented.

## 2. Experimental

### 2.1. Materials

A commercial batch of benzocaine (purity ≥ 99%, Sigma-Aldrich, Saint-QuentinFallavier, France) was recrystallised in ethanol. The obtained crystals (Figure 2) were taken out of the solution and left to dry completely under vacuum. Verification via X-ray diffraction demonstrated that only benzocaine form I was present in the recrystallised batch. After drying under vacuum, the recrystallised sample was used without further purification.

### 2.2. Adiabatic Calorimetry

The sample holder of the adiabatic calorimeter was filled with a quantity of 2.8650 g of benzocaine form I crystals, weighed on a precision balance. The description of the calorimeter has been published previously [24]. Once form I was loaded, it was cooled down to 100 K to measure the heat capacity to room temperature. No phase transition was observed. Subsequently, heat capacity measurements from about 10 K with liquid helium were carried out up until the transition from form I into form II. After this transition, the sample was cooled down again to 100 K to measure the heat capacity and observe the fully reverting form III to form II transition (form III is obtained at about 260 K by cooling form II [1]). In the final stage, the sample was cooled down to about 10 K and the heat capacity of forms III and II up until their melting and into the liquid state was obtained.

### 2.3. Differential Scanning Calorimetry

Differential scanning calorimetry (DSC) experiments were carried out with a conventional Q100 thermal analyser from TA Instruments. It was calibrated using the melting point of indium (*T*_fus_ = 429.75 K and ∆_fus_*H* = 3.267 kJ mol^−1^). The specimens with sample masses around 10 mg were weighed with a microbalance sensitive to 0.01 mg, and sealed in aluminium pans. DSC runs were carried out with a heating rate of 2 K min^−1^.

In addition, a PYRIS Diamond DSC from Perkin Elmer, which uses the power compensation technique, was used for differential scanning calorimetry measurements. It was calibrated using the melting point of indium (details above) and the solid–solid phase transition of cyclohexane (*T*_II-I_ = 186.09 K and ∆_II-I_*H* = 6.686 kJ mol^−1^). The samples were sealed in aluminium pans and runs were carried out with a heating rate of 5 K min^−1^.

### 2.4. Powder X-ray Diffraction

High-resolution X-ray diffraction as a function of temperature was performed at a standard pressure using Debye-Scherrer geometry and the transmission mode. CuKα_1_ (λ = 1.54056 Å) radiation was used in a horizontally mounted INEL diffractometer with a graphite monochromator and a cylindrical position-sensitive detector (CPS-120) with 4096 channels (0.029° 2*θ*-angular step). Powder samples were introduced into a 0.5 mm diameter Lindemann capillary. The temperature was controlled with a 700 series Oxford Cryostream Cooler from Oxford UK Cryosystems.

Lattice parameters as a function of temperature were determined through the isothermal acquisition of X-ray patterns between 120 K and 355 K, just below the melting point of benzocaine. Acquisitions took at least 1 h after a temperature stabilization of 15 min, while the heating rate between the measurements was 1.3 K min^−1^. External calibration by the means of cubic phase Na_2_Ca_3_Al_2_F_4_ and cubic spline fitting was used to convert the measurement channels into 2*θ*. The peak positions were determined using pseudo-Voigt fitting and the unit cell parameters were obtained from these fits using Fullprof and Topas Academic v4.1 [25,26,27].

### 2.5. High-Pressure Differential Thermal Analysis

High-pressure differential thermal analysis (HP-DTA) measurements were carried out with an in-house-constructed apparatus similar to the equipment built by Würflinger [28], operating between 0 and 300 MPa. Solid–solid transition temperatures between the different polymorphs of benzocaine and the melting temperatures as a function of pressure were determined for specimens mixed with an inert perfluorinated liquid (Galden; Bioblock Scientifics, Illkirch, France) to eliminate air while enclosed in tin capsules. The onset of the calorimetric peaks was taken as the transition temperature.

## 3. Results

### 3.1. The I→ II Transformation

The recrystallisation of benzocaine from ethanol leads to form I (P2_1_/c), as testified by the resulting diffraction pattern in Figure 3 at 295 K. On heating, form I gradually converts into form II, which is the only form present at 353 K according to the data in Figure 3.

From the DSC data in Table 2 and the DSC curve in Appendix A, it can be seen that the transformation of form I into form II is a slow and not very well controlled process. In many cases, two peaks are observed (Table 2 and Appendix A) and the enthalpy varies considerably. Nonetheless, the peaks can most likely be ascribed to the I → II transition based on the observations of the X-ray diffraction in Figure 3. Considering that the peaks in the DSC curve around 317 and 340 K are both related to the I → II transition, a clear temperature for the phase equilibrium cannot be given. The average combined enthalpy amounts to 1.6 J g^−1^ for the TA instruments and 2.0 J g^−1^ for the Perkin Elmer. Averaging over both machines leads to a value of 1.8 J g^−1^, although the variability most likely indicates that the enthalpy of the transition is mostly underestimated and the real transition enthalpy may be close or even above the highest observed value of 3.0 J g^−1^ (the combined transition of sample 8 in Table 2). The melting enthalpy difference and temperature averaged over all measurements are 132(3) J g^−1^ and 362.1(1.0) K (88.9 °C), respectively.

### 3.2. The Thermal Expansion of Benzocaine

The unit cell parameters of form I as a function of temperature have been compiled in Appendix A (see Section 2.4 for the measurement procedure). It leads to the following expression for the specific volume of form I as a function of temperature obtained between 120 K and 320 K:v_I_(*T*)/cm^3^ g^−1^ = 0.788(3) + 3.8(2.2) × 10^−5^ *T*/K + 3.0(5) × 10^−7^ *T*^2^/K (3)

Within the fitting range, the average deviation of this expression in relation to the measured data is less than 0.1%.

To ensure that the volume differences between the polymorphs are consistent, the diffraction of the other two polymorphs has been measured with the same experimental settings and with the same sample as form I. These specific volumes match with the error of the previously obtained data [1]. The unit cell parameters of forms II and III as a function of temperature can be found in Appendix A. These data lead to the following expressions for the specific volume of forms II and III of benzocaine:v_II_(*T*)/cm^3^g^−1^ = 0.835(30) − 2.19(1.99) × 10^−4^ *T*/K + 7.15(3.27) × 10^−7^ *T*/K (4)
v_III_(*T*)/cm^3^g^−1^ = 0.798(3) − 2.78(2.85) × 10^−5^ *T*/K + 4.91(74) × 10^−7^ *T*/K(5)

The average error of these fits within their fitting ranges is 0.13% (form II, 250–355 K) and 0.05% (form III, 120–255 K). Moreover, it can be seen in Figure 4 that form I is the densest form (Equation (3)) as it has the lowest specific volume from the lowest measured temperature to its equilibrium with form II. Form III has a slightly lower density (Equation (5)), and the least dense form is form II (Equation (4)).

### 3.3. Heat Capacity, Entropy, Enthalpy, and Gibbs Free Energy Obtained via Adiabatic Calorimetry

The heat capacity of form I has been measured by adiabatic calorimetry from 12.6 K onwards, while the heat capacity below that temperature has been obtained by extrapolation with Appendix A (see Figure 5 and Appendix A). Form I turns into form II at 319.5 K, as indicated by the relatively broad peak (red line in Figure 5), which compares with the slow conversion of form I into form II as observed by DSC. The entropy and enthalpy involved with this transformation and obtained via adiabatic calorimetry can be found in Table 3.

Once converted into form II, the heat capacity was once again measured from 11.4 K onwards, with an extrapolation to 0 K by Appendix A. The data can be found in Appendix A. Upon cooling, form II turns into form III at about 260 K depending on the cooling rate, which is the actual polymorph measured at a low temperature. A peak is observed (blue line in Figure 5) at 266.0 K, indicating the transition from form III to form II, which then melts at 361.0 K. The entropy and enthalpy changes of the two transitions obtained via adiabatic calorimetry have been compiled in Table 3.

### 3.4. Transition Temperatures Obtained under Pressure

The analysis of the pressure–temperature phase diagram, which will follow in the discussion, led to the realisation that the I–II and I–L phase equilibria were accessible for measurement in the high-pressure differential thermal analyser. Hence, the samples of form I have been loaded in tin capsules and measurements as a function of temperature for a given pressure have been carried out. Moreover, further analysis of the phase diagram demonstrated that form I could be regained through crystallising the melt at high pressure and cooling down. The peaks indicating that form I is obtained again can be observed in Figure 6.

The transition pressures as a function of temperature of the two-phase equilibria I–II, II-L and I–L, obtained via HP-DTA in the current study have been compiled in Appendix A. They result in the following equations of the transition pressure (P/MPa) fitted to the data:*P*_I-II_/MPa = −598(22) + 1.89(7) *T*/K   R^2^ = 0.992 (6)
*P*_II-L_/MPa = −2350(128) + 6.5(4) *T*/K   R^2^ = 0.98 (7)
*P*_I-L_/MPa = −2131(121) + 5.9(3) *T*/K   R^2^ = 0.989(8)

## 4. Discussion

### 4.1. Stability of Benzocaine under Ordinary Pressure: Adiabatic versus DSC

The stability of the different polymorphs can be directly obtained from the heat capacity data, which have been converted to entropy, enthalpy, and Gibbs free energy in Appendix A. The Gibbs free energy of form I has been set to zero at 0 K, resulting in a positive Gibbs free energy of 197.4 J mol^−1^ for form III. The Gibbs free energies of forms III and II meet at 266 K and there should be a slight change in the slope in the blue curve in Figure 7. The Gibbs free energy of form I meets that of form II at 319.5 K, where both the green curve and the blue curve in Figure 7 become identical. A final change in the slope of the blue curve should occur at 361.0 K, where form II is replaced by the liquid as the most stable phase.

Comparison between Table 2 and Table 3 demonstrates that the I–II transition possesses indeed a higher transition enthalpy than obtained with DSC, which is probably due to the fact that part of the solid–solid transition is not fully observed by DSC, because the signal may be spread over a larger temperature domain and fade into the baseline. This highlights the importance of adiabatic calorimetry, which clearly provides better equilibrium data for slow solid–solid transitions. The melting enthalpy obtained with the recent batches is slightly smaller for the values obtained with DSC. Although the scatter is relatively large and within the measurement error, the DSC and adiabatic calorimetric melting data are the same. The previously published DSC data on forms II and III appear to be a little higher than the adiabatic calorimetric data. The reason for this is not clear. The calibration of the DSC will have been different, and a different batch of benzocaine was used.

### 4.2. Construction of the Pressure–Temperature Phase Diagram

#### 4.2.1. High-Pressure Data and Its Triple Points

In Figure 8, the high-pressure data (solid circles) and the fitted lines (Equations (1) and (6)–(8)) have been plotted. The previously measured II–L equilibrium (Equation (2)) has not been used, because the fit may have been based on the melting equilibria of both form I and form II. It can be seen that both II–L and I–L (Equations (7) and (8), respectively) are very close to the former equilibrium Equation (2). From this experimental phase diagram, it is immediately clear that at 0 MPa, form II melts at the highest temperature (purple line) and that a stable I-II equilibrium occurs slightly above 300 K. The blue–purple triangle that is formed with the dashed 0 MPa baseline represents the small P–T domain in which form II is stable. Form I is stable on the upper left-hand side of the blue line and the III–II equilibrium, the green line, must be metastable. Because form I is stable above the blue I–II equilibrium, the red dotted line, representing the melting of form I, becomes stable once it intersects the blue line towards higher pressures, while it is found at the right-hand side of the purple II–L equilibrium. The intersection of the blue, purple, and red lines is the stable I–II–L triple point, of which the coordinates are listed in Table 4 below. Moreover, the intersection of the green III–II and purple II–L equilibria provides the position of the III–II–L triple point. Finally, the green III–II and blue I–II equilibria intersect at a negative pressure at the I–II–III triple point. Each intersection of the four two-phase equilibria, with their respective vapour phase pressures, close to 0 MPa (and approximated by the dashed black line), represents a triple point involving the vapour phase, *v*, (Table 4, in order of increasing temperature): III–II–*v*, I–II–*v*, I–L-*v*, II–L–*v*. Here, the triple-point temperature of III–II–*v* is determined with Equation (1), which equals to 267.8 K at the MPa value of 0. For the triple point I–II–*v*, Equation (6) is used, leading to a temperature of 316.6 K. In the same way, the triple point of II–L–*v* is determined with Equation (8), leading to 359.5 K, while the triple point temperature II–L–*v* follows from Equation (7) and gives rise to 362.0 K. The calculations of the triple points have been carried out with the equations within which the parameters have not been rounded off, and the resulting triple-point temperatures may therefore differ slightly if the equations are used as listed in this paper.

The remaining question is the position of the I–III equilibrium and connected to this, the position of the III–L equilibrium. This will be solved in a topological manner, as discussed in the next section.

#### 4.2.2. The Positions of the I–III and the III–L Equilibria

The positions of the I–III and the III–L equilibria in the phase diagram are closely linked. It is, however, easiest to locate the I–III equilibrium, because its position has more dramatic consequences for the phase behaviour of benzocaine. In Figure 8, there is one triple point, I–II–III, at a low temperature and negative pressure, which must be intersected by the I–III equilibrium. From Equations (3) and (5) and Figure 4, it can be seen that form I has a smaller specific volume, thus form I is the stable high-pressure form in relation to form III according to Le Chatelier. The differences in entropy of the polymorphs I and III in relation to form II are listed in Table 3. The entropy change going from form III to form II is smaller than the entropy change for form I to form II. This implies that form III contains more entropy and is therefore the high-temperature form in relation to form I. The same conclusion can be reached by comparing the entropies listed in Appendix A. Thus, simply considering the global inequalities in the volume and entropy, it is found that in relation to the I–III equilibrium, form I is the low-temperature, high-pressure form and form III is the high-temperature, low-pressure form; in other words, the phase equilibrium between the two polymorphs has a positive slope with form I on the upper left-hand side and form III on the lower right-hand side.

Taking the I–II–III triple point (T = 154 K and P = −307 MPa) as a pivot for the I–III equilibrium, and taking into consideration its positive slope, three topological scenarios exist: (1) I–III with a steeper slope than III–II and I–II, (2) I–III with a less steep slope than III–II, but a steeper slope than I–II, and (3) I–III with a shallower slope than both other solid–solid equilibria. It should be kept in mind that form I is stable above the equilibrium and form III below it. In the case of scenario (1), form I will be stable at high pressure and low temperature. Form III will appear on crossing this equilibrium, which will then turn into form II on crossing from the top–left III–II equilibrium; however, at that point, form II is still metastable with respect to form I, and that implies that form III is also still metastable with respect to form I, which creates an inconsistency in the phase behaviour. A similar inconsistency exists for scenario (2): form I, which is stable according to the phase diagram in Figure 8, turns into form III, which must be metastable, because form II is the more stable form. This implies that form I must be metastable towards form II, but that only occurs once the blue I–II equilibrium is crossed. Thus, the only viable scenario is number (3), for which the slope of the I–III equilibrium is the smallest of the three solid–solid equilibria and the line is therefore found below the I–II phase equilibrium. This also implies that the I–III equilibrium is completely metastable, because neither form I nor form III are stable below the I–II equilibrium, implying that Figure 8 already covers the full stable phase diagram. The remaining unknown is the slope of the I–III equilibrium and where it precisely intersects the vapour phase pressure close to 0 MPa, which would represent the I–III transition under ordinary conditions (i.e., in the calorimeter).

There are several approaches to estimate the slope of the I–III equilibrium, of which the following two are the most direct: (1) by calculating the slope using the Clapeyron equation, or (2) by calculating the I–III–*v* (*v* = vapour) triple point, the transition temperature at a standard pressure (0 MPa), using an equation proposed by Yu [29]. With option (1), the slope with the I–II–III triple point will define the position of the equilibrium expressed as a straight line, and with option (2), a straight line drawn through the two triple points I–II–III and I–III–*v* will define the position of the equilibrium. Because both are abstract extrapolations, neither will necessarily represent the realistically precise position of the equilibrium, but topologically, it will provide the correct interpretation of the phase behaviour, which should be thermodynamically consistent.

Using option (1), the calculation of the slope necessitates the Clapeyron equation:(9)$$\frac{dP}{dT}=\frac{\Delta S}{\Delta v}$$
in which *dP*/*dT* is the slope in the pressure–temperature phase diagram, ∆*S* is the entropy difference and ∆*v* is the volume difference between the two phases, I and III. ∆*S* follows from Table 3 (or the Appendix A) and is ∆_I→III_*S* = ∆_I→II_*S* − ∆_III→II_*S* = 0.011938 − 0.010225 = 0.001713 J g^−1^ K^−1^. In an ideal case for the difference in volume, the volume at the transition temperature under standard conditions should be taken, but this value is unknown and located at a higher temperature than that where forms III and I can be obtained. Therefore, it is more convenient to choose a temperature at which both volumes are accurately defined, such as 225 K, using the Equations (3) and (5). This leads to *v*_I_ = 0.81221 cm^3^ g^−1^ and *v*_III_ = 0.81613 cm^3^ g^−1^, and thus to a difference of ∆_I→III_*v* = *v*_III_ − *v*_I_ = 0.00392 cm^3^ g^−1^. Using the Clapeyron equation Equation (9), this leads to a slope of 0.4375 MPa K^−1^, and using the triple point I–II–III (Table 4), a tentative equation for the I–III equilibrium is obtained: *P*_I-III_/MPa = −374 + 0.437 *T*/K. It can be seen that the slope is indeed much less than that of the III–II equilibrium (Equation (1)), with 2.70 MPa K^−1^ or the I–II equilibrium with 1.89 MPa K^−1^ (Equation (6)). Through this equation for the I–III equilibrium, the triple-point temperature of I–III–*v* is found to be 856 K.

Using option (2) requires an equation proposed by Yu to calculate the solid–solid transition temperature [29]:(10)$$T_{I\to \mathrm{III}}=\frac{\Delta_{\mathrm{III}\to L}H-\Delta_{I\to L}H}{\Delta_{\mathrm{III}\to L}S-\Delta_{I\to L}S}=\frac{\Delta_{\mathrm{II}\to L}H+\Delta_{\mathrm{III}\to \mathrm{II}}H-(\Delta_{\mathrm{II}\to L}H+\Delta_{I\to \mathrm{II}}H)}{\Delta_{\mathrm{II}\to L}S+\Delta_{\mathrm{III}\to \mathrm{II}}S-(\Delta_{\mathrm{II}\to L}S+\Delta_{I\to \mathrm{II}}S)}=\frac{\Delta_{\mathrm{III}\to \mathrm{II}}H-\Delta_{I\to \mathrm{II}}H}{\Delta_{\mathrm{III}\to \mathrm{II}}S-\Delta_{I\to \mathrm{II}}S}$$

As most of the transition enthalpies and entropies are not known, because only the fusion of form II has been measured, Equation (10) has been rewritten to provide the transition temperature between forms I and III using form II instead of the liquid phase, which is thermodynamically completely equivalent to the proposed equation by Yu involving the liquid phase. This leads to a temperature for the I–III–*v* triple point of 639 K. As this is at “ordinary pressure”, the pressure value will be approximated with 0 MPa. Then, with the I–II–III triple point at 154 K and −307 MPa, the following equation for the I–III equilibrium can be obtained: *P*_I-III_/MPa = −404 + 0.633 *T*/K. Again, it can be seen that the slope is rather small and much less than the slopes of the I–II and II–III equilibria, and the two scenarios can be considered to roughly represent the error margin of the I–III equilibrium, which will always be metastable and out of reach of measurement.

To draw the topological phase diagram, the latter expression will be chosen (there is no scientific ground to choose one expression over the other):*P*_I-III_/MPa = −404 + 0.633 *T*/K (11)

Once the position of the I–III equilibrium has been defined, it also defines the position of the melting equilibrium of form III, because the I–III equilibrium will intersect the I–L equilibrium, leading to the triple point I–III–L. Thus, with Equations (8) and (11), the following coordinates are found: 326 K and −198 MPa. Another triple point of the III–L equilibrium is III–II–L, which can be calculated with the measurement Equations (1) and (7). The triple-point coordinates are 429 K and 435 MPa. Therefore, the III–L equilibrium passing through these two points possesses the following equation:*P*_III-L_/MPa = −2204 + 6.15 *T*/K (12)

Using this equation, the melting point of form III can be determined, as this will occur at 0 MPa (ordinary pressure), leading to 358.3 K. This value, obtained through the extrapolation of the topological phase diagram, can be compared with the melting point that would be obtained by the equation proposed by Yu, and then modified to obtain a melting transition [1]:(13)$$T_{\mathrm{III}\to L}=\frac{\Delta_{\mathrm{II}\to L}H+\Delta_{\mathrm{III}\to \mathrm{II}}H}{\frac{\Delta_{\mathrm{II}\to L}H}{T_{\mathrm{II}\to L}}+\frac{\Delta_{\mathrm{III}\to \mathrm{II}}H}{T_{\mathrm{III}\to \mathrm{II}}}}=\frac{\Delta_{\mathrm{II}\to L}H+\Delta_{\mathrm{III}\to \mathrm{II}}H}{\Delta_{\mathrm{II}\to L}S+\Delta_{\mathrm{III}\to \mathrm{II}}S}$$

Using the values from Table 3, this leads to the melting point of form III of 358.5 K, so the differences in this case are negligible. It can also be concluded that form III is the lowest-melting form, followed by form I, while form II is the highest-melting form (Table 4). Another observation that can be made is that the slope of the II–L equilibrium, 6.5 MPa K^−1^, is the steepest slope, while III–L has a slightly shallower slope of 6.15 MPa K^−1^ and form I has the gentlest slope with 5.9 MPa K^−1^. This sequence in the slopes leads to a stable melting equilibrium for form I at high pressure, whereas the melting equilibrium of form III is placed in such a way that it never becomes stable, reflecting the metastability of form III.

#### 4.2.3. The Pressure–Temperature Phase Diagram

In a pressure–temperature phase diagram with three solid phases, the liquid and the vapour phase, ten triple points must be present, which represent the ten possible intersections between the six two-phase equilibria [30]. All triple points have been listed in Table 4, based on the measured phase equilibrium data with the high-pressure differential thermal analysis. Only, when necessary, as in the case of the slope of the I–III equilibrium enthalpy and entropy values obtained via adiabatic calorimetry have been used. Therefore, there are small differences between the transition temperatures obtained via HP-DTA and DSC or adiabatic calorimetry, which can be considered the uncertainty over the measurements. Moreover, it is possible to calculate the vapour pressure of the condensed phases using the Clausius–Clapeyron equation, and the boiling temperature and enthalpy of vaporization obtained through ACD-Labs [31], as it has previously been carried out for the equilibria involving the vapour phase and forms II and III [1]. This approach is explained in detail in the Appendix A. The calculated vapour pressures of the respective triple points have been compiled in Table 4 (see also Appendix A).

The resulting topological phase diagram is presented in Figure 9 on the left-hand side. For the sake of clarity, the melting equilibria (triple points 1, 2, and 3) have been separated in an exaggerated way. The stable vapour-phase equilibria defining the lower limit of the stable condensed phases are given by the sequence f-7-1-a. The stability domain of form II is defined by the black triangle 1-4-7, and above this triangle f-7-4-c, defines the stability domain of form I. Form III becomes more stable than form I below the equilibrium line indicated by ‘j’, but its position is such that the vapour phase has the lowest Gibbs free energy and is thus the more stable phase. The stable phase diagram of benzocaine with the phases I, II, liquid, and vapour, can be found in the right-hand panel of Figure 9. The temperature and pressure axes are to scale.

Although the phase diagram involving three phases seems rather complicated, it should be realised that it consists of a sum of three dimorphism phase diagrams, the first being the previously published III–II phase diagram, where both form III and form II have a stable temperature domain at a standard pressure (enantiotropy), whereas the system becomes monotropic at higher pressures (see blue lines and point 5 in Figure 9) with form III, i.e., the single more stable form. This phase behaviour is similar to that of sulphur (Figure 1) [4]. The second phase diagram is that of forms I and II, which presents a very similar behaviour as forms II and III, except that now, form I is the high-pressure form (see black lines in Figure 9). As both form I and form III compete for the high-pressure, low-temperature domain, the third phase diagram, that of forms I and III, demonstrates that form III is less stable than form I. In fact, when form III finally has a lower Gibbs energy than form I, it is always the vapour that possesses an even lower Gibbs energy. The I–III equilibrium is indicated by the line j–j, which is mostly green (hypermetastable). Due to the overall monotropy between forms I and III, form III does not possess any stable pressure–temperature domain.

The pressure–temperature phase diagrams related to dimorphism are relatively easy to understand, because they only contain four triple points, and therefore only four possible cases exists, which have been described by Bakhuis Roozeboom [4]. These four cases are overall monotropy (benzocaine I–III), overall enantiotropy, as in the case of gestodene [32], enantiotropy turning monotropic upon increasing pressure (benzocaine II–III and I–II) and the inverse case, monotropy turning enantiotropic upon increasing pressure, of which ritonavir is an example [22]. The four resulting pressure–temperature phase diagrams can be found in a recently published paper [5].

Trimorphism, shown in Figure 9, becomes rapidly more complicated to describe than dimorphism, and in the literature, only four other papers report on topological phase diagrams involving trimorphism [33,34,35,36] and one other concerning tetramorphism [13]. In the case of the trimorphism of ferrocene [33], one of the phases is also overall metastable. As in the case of sulphur [4], the two other phases of ferrocene are overall enantiotropic, exhibiting stable domains irrespective of the pressure. For piracetam, all three forms have their stable domain, but only one remains stable under higher pressure, i.e., the system becomes fully monotropic [34]. In the case of 2-methyl-2-chloropropane, two solid phases are overall enantiotropic, whereas the third one becomes monotropic at a high pressure [36]. In the case of l-tyrosine ethyl ester, again, a sulphur-type phase behaviour between the two phases is encountered, whereas the third phase appears to exhibit a stable domain at an extremely low temperature, so all three phases are stable in their respective temperature domain at low pressure [35]. A similar line-up of stability domains as a function of temperature is observed for tetramorphic pyrazinamide, while under higher pressure, only two polymorphs remain [13]. Thus, even within this small collection of samples, many different combinations of phase behaviour are encountered.

## 5. Conclusions

Adiabatic calorimetry and high-pressure differential thermal analysis both lead to the conclusion that benzocaine form I possesses the largest stability domain, and that form II is only stable in a small pressure–temperature domain before melting. Full consistency exists between the outcomes of the two methods. A combination of the adiabatic calorimetry with powder X-ray diffraction for the specific volumes and the high-pressure measurements has led to a complete topological pressure–temperature phase diagram (Figure 9), in which it is demonstrated that form III does not possess any stable domain (Figure 9 and Appendix A). This case is a prime example of how phase behaviour can be added up and that the relative hierarchy between the phases does not change, while the absolute hierarchy is adapted as a function of the stability behaviour of a new phase. Form IV [14], for example, can be added to this phase diagram, once more information about its phase relationship with form I is obtained. It should also be clear that adiabatic calorimetry is much more reliable in finding equilibrium conditions than differential scanning calorimetry (DSC).

Even though form I is not the commercially available polymorph, adiabatic calorimetry has demonstrated that it is the most stable form from 0 K to *T*_I-II_ = 319.5 K. Nonetheless, form II is the commercial form, and it does not appear to transform at all into form I under standard conditions at room temperature, despite its metastable character, with respect to form I. The only way to obtain form I appears to be through the crystallisation of ethanol. However, while constructing the topological phase diagram of benzocaine, it became clear that the I–II–L triple point (point 4 in Figure 9) was in reach of the limits of the HP-DTA equipment, and that the melting equilibrium of form I could be observed. This has also led to the assumption that by melting form II, increasing the pressure, and lowering the temperature, once above the I–II–L triple point pressure, form I may crystallise instead of form II. The appearance of the I–II transition in subsequent runs confirmed this assumption (Figure 6).

Interestingly, form III, which appeared to be a low-temperature, high-pressure polymorph, at least with respect to form II [1], is in fact a high-temperature, low-pressure form in relation to the more stable form I. The reason that form III appears at all, is that once polymorph II has formed, it has never been observed to convert back into form I; therefore, commercial batches contain form II. Due to this persistence, form III can appear at low temperatures as a means for the system to lower its energy towards form I, even if the system remains trapped, as a direct transition between form I and form III has never been observed either.

For pharmaceutical applications, these findings imply that form II can be safely used for the development of solid-state formulations. This would render the formulations safe against high temperatures. The reason why polymorphs II and III are so persistent in relation to a more stable form should be investigated more thoroughly, so that a drug molecule may be designed with highly persistent metastable polymorphs, which leads to improved solubilization and bioavailability.

## Figures and Tables

**Figure 1 pharmaceutics-15-01549-f001:**
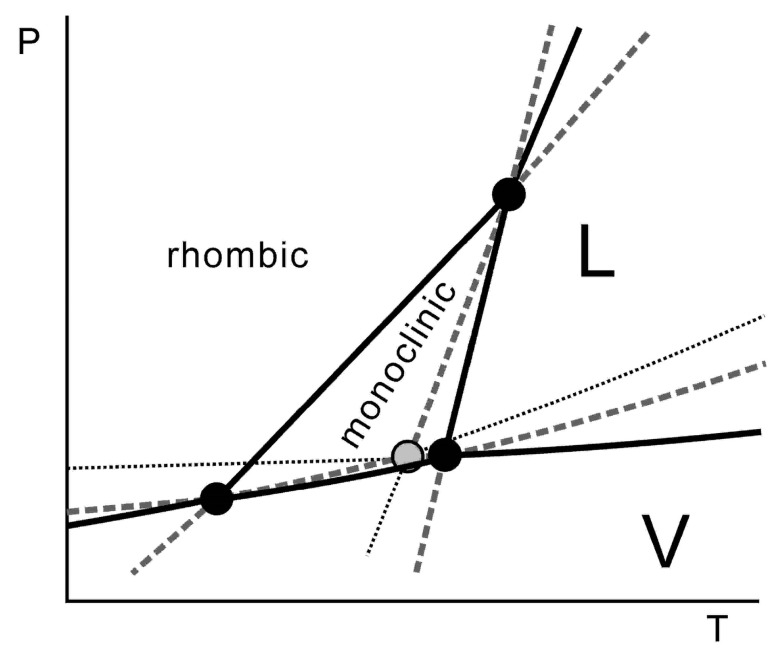
The pressure–temperature phase diagram of sulphur. Rhombic and monoclinic are the two polymorphs: L is the liquid and V is the vapour phase. Solid lines are stable, dashed lines are metastable, and dotted lines are supermetastable. At low pressures, both the rhombic and monoclinic forms of sulphur possess a stable domain. At a high pressure, only the rhombic form remains. This is an example of an enantiotropic system turning monotropic upon increasing pressure, as the stable domain of the monoclinic form ends at a given pressure. In the case of the previously published II–III phase diagram of benzocaine [1], the rhombic is replaced by polymorph III and monoclinic by polymorph II.

**Figure 2 pharmaceutics-15-01549-f002:**
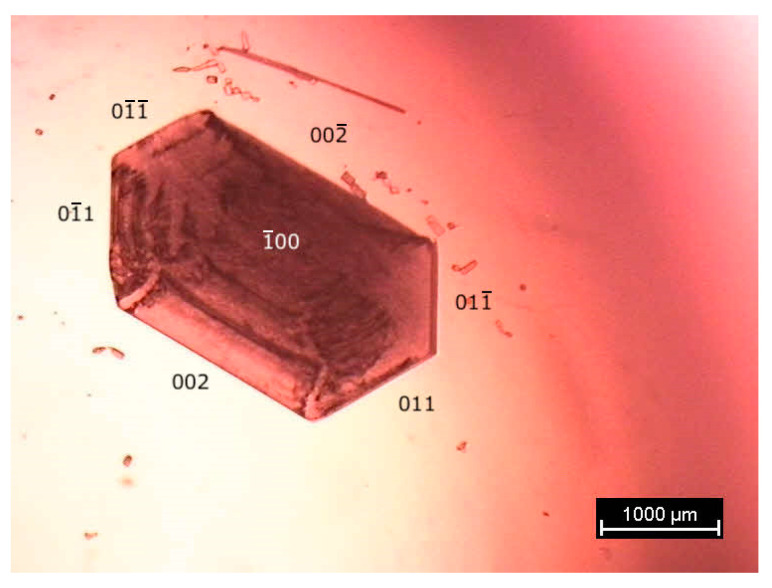
A photograph of a crystal of benzocaine (C_9_H_11_NO_2_, 165.19 g mol^−1^) form I obtained from an ethanol solution.

**Figure 3 pharmaceutics-15-01549-f003:**
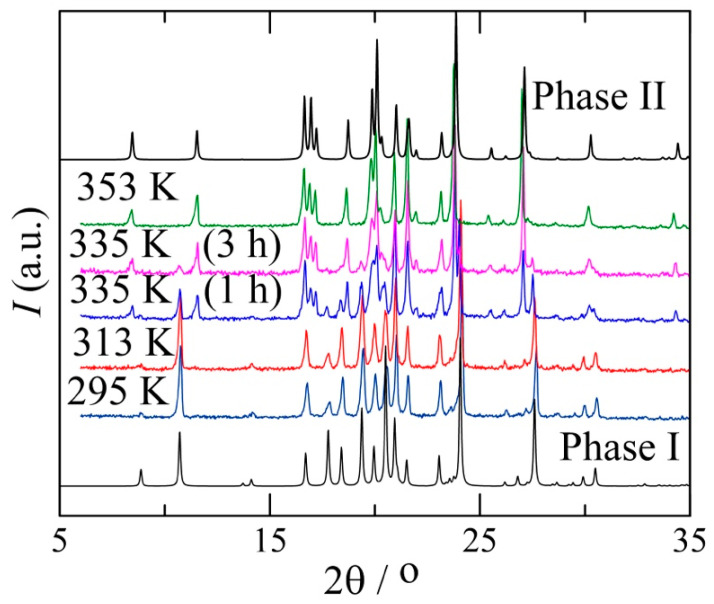
Benzocaine form I at 295 K (22 °C) after recrystallisation in ethanol. At 313 K (40 °C), form I still appears to be stable; however, further heating slowly leads to the formation of form II starting at 335 K (62 °C), as demonstrated by two measurements, 1 h each, at this temperature. Form II is clearly the only polymorph present at 353 K (80 °C). The diffraction patterns of phases I and II (in black) have been provided based on single-crystal diffraction data from the CSD.

**Figure 4 pharmaceutics-15-01549-f004:**
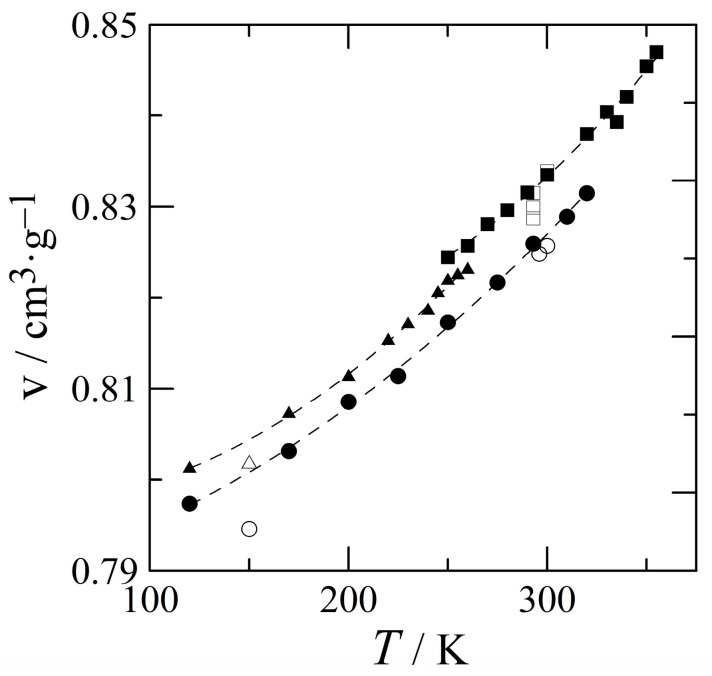
Specific volumes of forms I (filled circles), II (filled squares), and III (filled triangles) obtained by powder X-ray diffraction in the current study. Single-crystal literature data are indicated by open symbols.

**Figure 5 pharmaceutics-15-01549-f005:**
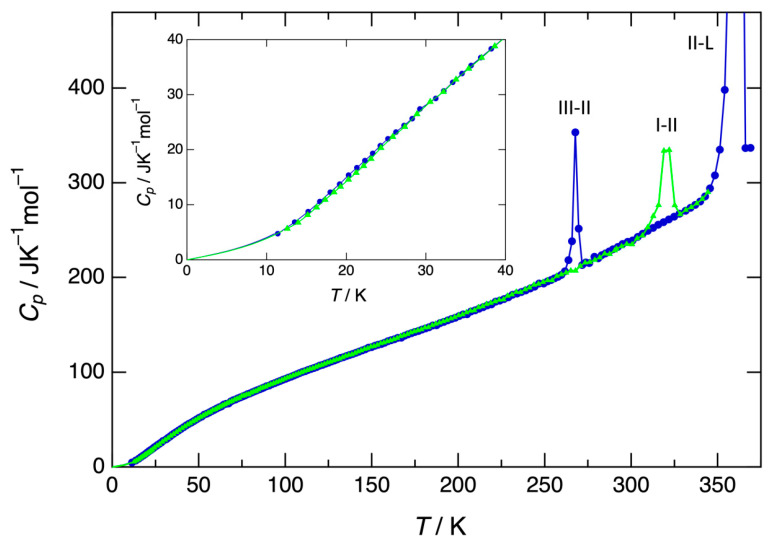
Heat capacity data of forms I (green triangles) and III (blue circles), turning into form II at the respective peaks, followed by the melting peak (in blue).

**Figure 6 pharmaceutics-15-01549-f006:**
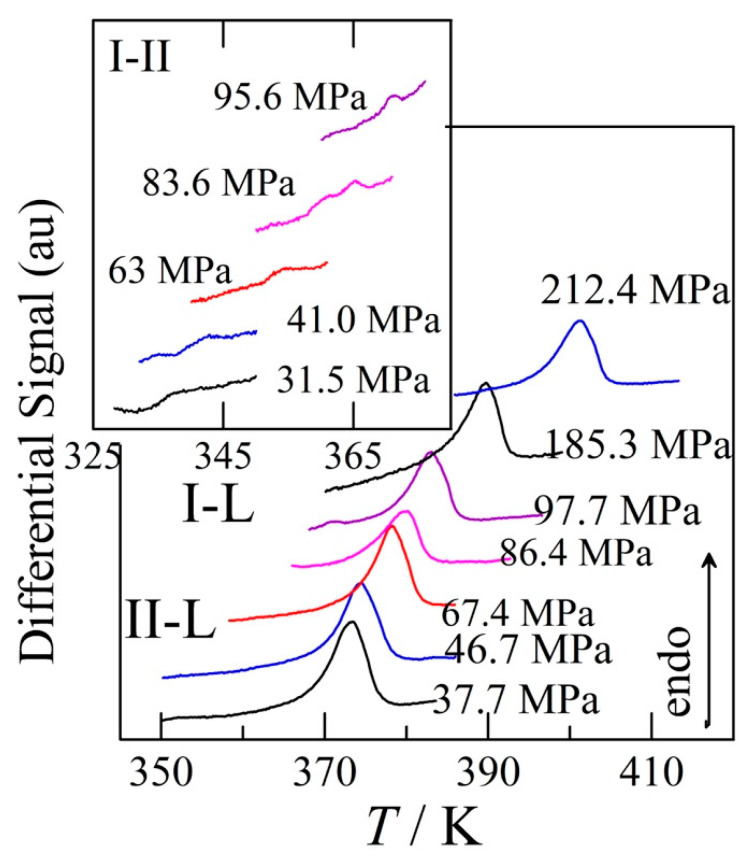
High-pressure differential thermal analysis of the I–II transition (inset), and the melting peaks of either form II (lower pressure) or form I (higher pressure).

**Figure 7 pharmaceutics-15-01549-f007:**
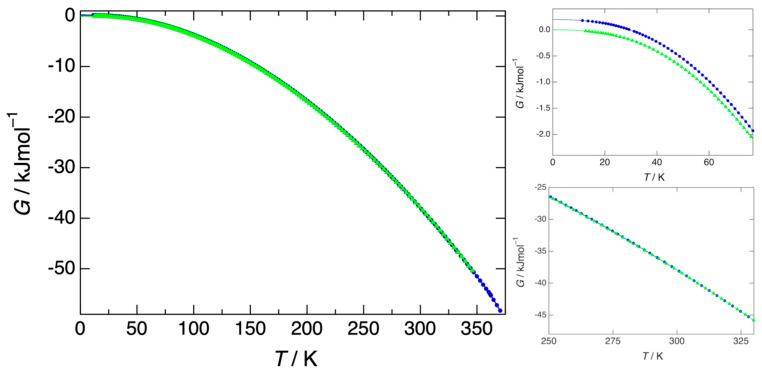
Gibbs free energy of form I (green triangles) and form III (blue circles), which become equal once both forms have turned into form II (bottom right-hand figure). Form III has a higher Gibbs free energy (top right-hand figure), and is therefore less stable than form I. Form I remains the most stable form to 319.5 K.

**Figure 8 pharmaceutics-15-01549-f008:**
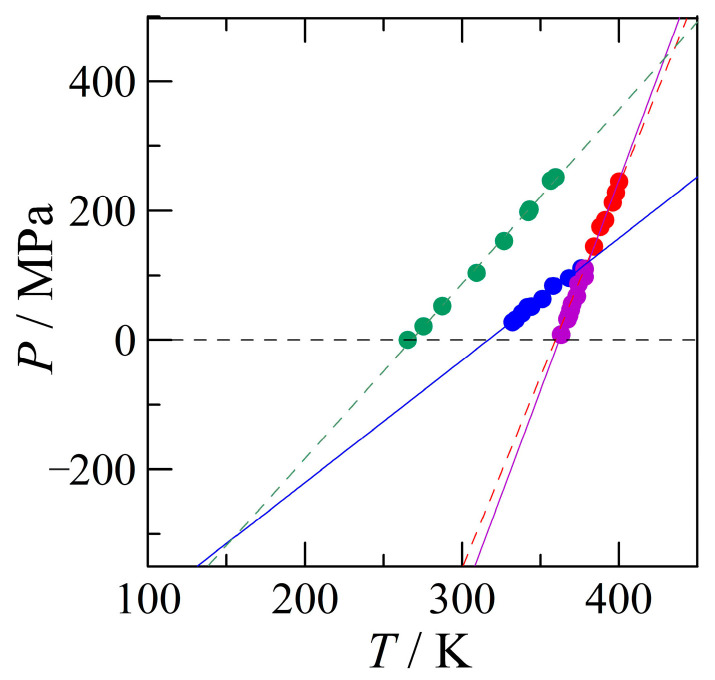
The pressure–temperature phase diagram based exclusively on the obtained HP-DTA data measured in the current study (equilibrium I–II dark blue, equilibrium II–L purple, equilibrium I–L red) and previously measured data (equilibrium III–II green) [1].

**Figure 9 pharmaceutics-15-01549-f009:**
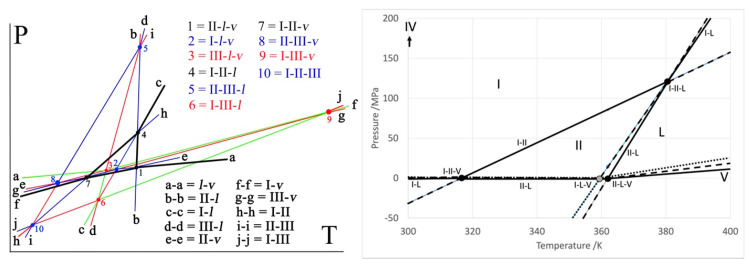
**Left-hand panel.** Topological phase diagram of the three polymorphs of benzocaine together with the liquid and vapour phases. The 10 triple points have been numbered and are listed in the graph, as are the different two-phase equilibrium lines. Black indicates stable, blue metastable, red supermetastable, and green hypermetastable. Form I is stable in the upper left-hand corner defined by the black lines, the liquid is stable in the upper right-hand corner defined by the black lines, the vapour is stable below the black lines, and form II is stable within the triangle formed by the black lines. A close-up of the triple points 1, 2, 3, 7, and 8 involving the vapour phase equilibria can be found in Appendix A. **Right-hand panel.** Stable phase diagram of benzocaine with the pressure and temperature axes to scale. It should be noted that the vapour pressure equilibria (lines marked on the left with e, f, and g) are now fully superposed. Form III, which does not possess a stable domain, is absent in this diagram. Form IV stabilizes somewhere above 0.5 GPa at room temperature. The stability hierarchy is indicated similarly to Figure 1: solid lines and circles are stable two-phase equilibria and stable triple points, respectively, the dashed lines and grey circles indicate metastable, and dotted lines indicate supermetastable.

**Table 1 pharmaceutics-15-01549-t001:** The crystal structures in the Cambridge Structural Database obtained at a normal pressure.

Code (QQQAXG)	Form	*T*/K	Space Group	*a*	*b*	*c*	*β*,*γ* ^b^	*V*_cell_/Å^3^	*v*_spec_/cm^3^ g^−1^	Ref.
02	I	150	*P*2_1_/*c*	8.198	5.430	19.592	91.35	871.9	0.7946	[15]
04	I	300	*P*2_1_/*c*	8.257	5.501	19.956	91.70	906.0	0.8257	[16]
06	I	300	*P*2_1_/*c*	8.257	5.501	19.956	91.70	906.0	0.8257	[17]
09	I	296	*P*2_1_/*c*	8.250	5.501	19.950	91.73	905.0	0.8248	[18]
00	II	293	*P*2_1_2_1_2_1_	5.307	8.222	20.869		910.6	0.8299	[19]
01	II	293	*P*2_1_2_1_2_1_	5.302	8.217	20.870		909.2	0.8287	[20]
05	II	300	*P*2_1_2_1_2_1_	5.311	8.242	20.904		915.0	0.8339	[16]
07	II	300	*P*2_1_2_1_2_1_	5.311	8.242	20.904		915.0	0.8339	[17]
10	II	293	*P*2_1_2_1_2_1_	5.309	8.236	20.866		912.4	0.8315	[18]
03	III	150	*P*2_1_	8.188	10.639	20.476	99.37	1759.9	0.8020	[16]
08	III	150	*P*2_1_	8.188	10.639	20.476	99.37	1759.9	0.8020	[17]
13	IV ^a^	298	*P*2_1_/*c*	6.305	5.184	24.940	96.25	810.3	0.7385	[14]

^a^ Only observed under pressure; for the listed unit cell parameters 0.55 GPa. ^b^ *β* for forms I and IV, and *γ* for form III.

**Table 2 pharmaceutics-15-01549-t002:** The transformation of form I into form II and fusion of benzocaine form II, as obtained by two differential scanning calorimeters (see also Appendix A).

		Peaks I → II				Fusion	
Sample	Mass/mg	*T*/K	∆*H*/J g^−1^	*T*/K	∆*H*/J g^−1^	*T*/K	∆*H*/J g^−1^
TA instruments							
1	3.84	316.8	0.67	340.2	1.09	362.6	132.9
2	3.11			341.6	0.89	362.6	137.5
3	7.89	317.0	1.70			362.8	127.4
4	1.66			340.0	1.86	362.6	129.9
5	2.42			342.4	2.63	362.6	134.4
6	3.70	318.0	0.03	344.2	1.54	362.6	133.5
7	4.44	317.6	0.50	339.2	0.61	362.6	131.0
Perkin Elmer							
8	2.46	321.0	0.76	339.6	2.24	359.8	129.9
9	2.08	316.2	0.62	332.6	1.97	361.4	127.5
10	2.15			336.2	0.90	361.8	131.5
11	2.42			343.8	1.67	361.4	133.1

**Table 3 pharmaceutics-15-01549-t003:** Temperature, entropy, and enthalpy of transition obtained with adiabatic calorimetry.

Equilibrium	*T* /K	∆*S* /J g^−1^ K^−1^	∆*H* /J g^−1^
III-II	266.0	0.010225	2.7194
I-II	319.5	0.011938	3.8146
II-L	361.0	0.369514	133.412

**Table 4 pharmaceutics-15-01549-t004:** Triple points involving polymorphs I, II, and III of benzocaine.

Triple Point	Temperature/K	Pressure/Pa	
I–II–III	154.0	−307 × 10^6^	metastable
III–II–*v*	267.8	0.012	metastable
I–II–*v*	316.6	2.4	stable
I–III–L	326.1	−198 × 10^6^	supermetastable
III–L–*v*	358.3	78	supermetastable
I–L–*v*	359.5	83	metastable
II–L–*v*	362.0	95	stable
I–II–L	380.1	120 × 10^6^	stable
III–II–L	428.9	435 × 10^6^	metastable
I–III–*v*	639.3	7.4 × 10^6^	supermetastable

## Data Availability

Most data is available in the paper or in the Appendix A. Any additional information can be obtained upon request.

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
