# Peer review of "The Phase Diagram of the API Benzocaine and Its Highly Persistent, Metastable Crystalline Polymorphs"

_pharmaceutics, 2023, doi:10.3390/pharmaceutics15051549_

Round 1

Reviewer 1 Report

This manuscript deals with the phase diagram of the API benzocaine and its highly persistent, metastable crystalline polymorphs are described.  

However, the manuscript is very complicated and totally the explanation for the diagram is not sufficient and not clear.

In DSC measurement, the temperature and the enthalpy for the transformation and the heat of fusion of the polymorphs are usually dispersed, and their values depend on the experimental conditions such as the increasing temperature. Is the accuracy enough for the diagram?

Fig 8 is so much complicated and unclear.  At least one or two enlarged figures for the typical system should be used and the stable and metastable zone of each polymorph should be shown to explain clearly.

 In Table 4: What the words of the metastable and supermetastable mean for the triple points? 

Reviewer 2 Report

This manuscript describes the stability relationships between form I and the other two forms II and III of API benzocaine by using adiabatic calorimetry, high-pressure differential thermal analysis and powder X-ray diffraction.  This study is worthy of publication from a viewpoint of development of desired polymorphic forms of benzocaine.  The greater part of this manuscript is devoted only to mention the data interpretation for the phase diagram completion.  The phenomenological and kinetic consideration, which is essential to get a good impression of the obtained results, is not enough. The referee has the following comments.

 (1) Equilibrium I-II

The author should mention the reason why polymorph II has formed and has never been observed to convert back into form I by using phase diagram.  The kinetic processes should be discussed separately from equilibrium theory.

 (2) Preparation of form I

The crystal purity of benzocaine form I obtained by recrystallization should be mentioned.  And the purity of crystal form should be also mentioned.

 (3) Recrystallization from ethanol.

The author should discuss why benzocaine form I can be obtained from ethanol solutions from the viewpoint of phase equilibrium.

 (4) Kinetic consideration

According to section 3.1, the transformation of form I into form II was discussed as kinetic process.  The validity of treating data obtained from kinetic process as phase equilibrium should be explained.

Reviewer 3 Report

Line 39 (page 2): you write that “Both forms convert into form II around room temperature”. Actually, III-II transition happens at about 266 K.

Line 50 (page 2): is it possible to add a table with the cell parameters of the four polymorphs?

Line 76 (page 4): the sentence “to observe the fully reversing form III to for II transition” is not clear. How do you obtain form III? In the previous sentence you speak about form I to form II transition.

Paragraph 2.3 (DSC): the measurements were performed in air? Which temperature variation rate was used?

Paragraph 2.4 (PXRD): Which temperature variation rate was used? Is it consistent with the one used during the DSC analysis? I think that considering the different value you found for the I->II transition by using different analytical techniques (320 or 340 K) you should try to increment temperature always (when possible) in the same way.   

Figure 2 (pag 6): I think that you should add to the figure the (theoretical) patterns of the two forms (I and II). In addition, I think that, due to the small dimensions of the figure, you could consider a smaller range of 2tetha (for example 5-35°?)  

Line 116 (page 6): I think that the speed of the I -> II phase transition could be better evaluated by using VT-PXRD instead of DSC data. Considering that DSC spectrum has an endothermic peak at about 310 K, have you tried,  during the variable temperature PXRD experiment, to leave your sample at that temperature (instead of 335 K as in Figure 2) to see if something change in the XRPD pattern?

Line 124 (page 6): can you add the DSC spectra?

Table 2 (Page 6/7): the values of enthalpy you found for the phase transition are, in my opinion, very low and I’m not sure about how significant they are (for example in sample 6 you reported 0.03 J/g). Could it be because phases I and II have very similar cell parameters and that the phase transition is a “simple” rearrangement of them? Perhaps a solid-state analysis of the crystal packing changes could help.

QQQAXG05: phase II, a= 8.242(1); b = 5.311(1); c = 20.904(1), beta = 90.

QQQZXG04: phase I, a = 8.257(1); b = 5.501(1); c = 19.956(2), beta = 91.70(1)

The sample you use during the DSC analysis are all from the same batch? I think that the presence of two endothermic peaks (something both present) must be further investigated.  

Line 129 (page 7): how did you determine the unit-cell parameters? Single crystal or powder X-ray diffraction? Add information about the procedure in the experimental chapter.

Line 163 (page 9): at which temperature II transforms in III during the cooling step?

Line 164 (page 10): between the III->II transition and the melting of II there are about 100 degrees (266->361K). At line 39 you write: “Both forms convert into form II around room temperature followed by melting”. This phrase seems indicate that the transition is followed (in few degree) by the melting of phase II. Perhaps you could modify the phrase?

Reviewer 4 Report

The phase diagram of the API benzocaine and its highly persis- 1
tent, metastable crystalline polymorphs 2
Ivo B. Rietveld1,2,3,*, Hiroshi Akiba1, Osamu Yamamuro1, Maria Barrio4, René Céolin4 and Josep-Lluis Tamarit4

p1) The latter two forms were known to have an enantiotropic phase relationship with 16
form III stable at low-temperature and high-pressure, while form II is stable at room temperature.
This sentence is inconsistent.  The latter  two phases are II and III.
so the phrase "relationship with form III" is inconsistent.
Moreover, it is better to cut this sentence in two pieces.

p2 l 31 enantiotropy turning into monotropy
for the non expert reader, it is advised to recall the definitions of these two words.

p2 l 47. form III. E value is not given

p3, l 54. recall once in English words the meaning of TII->I  and deltaH

equation (1) & (2).  The reader would appreciate to be given here also the temperatures
resulting in P_III-II = 0 and P_II-I  for these equations

Figure I. crystal photograph. can the indexation of main faces be given ?

Table 2. Authors should be more verbose in Table legends.
Explain why are there two  T/K and  deltaH columns under  Peaks I->II ?

Figure 3 legend.  Literature data of single crystal data ARE indicated by open symbols.

Figure is far from optimal.
In black & white hardcopy it does not look well.
Please use a dark and light color which can be differntiated in B&W.
Use two different symbols for the two curves, such as  circles and diamonds, for example.
The figure would gain in visibility if filled circle symbols were smaller.

p10 line 167.  I-I(italics) phase
please explain what the I_in_italics phase consists in
I have no idea what is it by reading the manuscript

Figure 6: same formatting remarks as in Fig 4
What is the uncertainty on measurement compared to the difference between two curves ?
add a zoom for the 300 - 380 K zone.
More marks should be put on the X axis.

4.2.2 The positions of the I-III and the III-L equilibria 233
The positions of the I-III and the III-l equilibria
is it  III-L or  III-I(italics) in the title ?

p13  l 235  because its position has more dramatic consequences

p 13  l237has the smaller volume ->  has a smaller specific volume ?

p20   so that 403
such behaviour may be programmed into a drug molecule through synthesis.
this phrase is not clear: rephrase.

/

Round 2

Reviewer 1 Report

I think this manuscript is improved and can be published.

Reviewer 2 Report

The referee confirmed the revision.

Reviewer 3 Report

I find the article sufficiently improved and I think that it can now be published on this journal.

Reviewer 4 Report

It would be better to use the word "liquid"  early in paragraph 4.2.2

to render the text more readily understandable for the reader.